# The effect of labour hopscotch framework on maternal and neonatal outcomes in pregnant women: A randomized controlled trial

Saeedeh Askari[1], Mina Iravani[1], Parvin Abedi[2]*, Bahman Cheraghian[3], Eesa Mohammadi[4], Shayesteh Jahanfar[5]

1 Department of Midwifery, Reproductive Health Promotion Research Center, Ahvaz Jundishapur University of Medical Sciences, Ahvaz, Iran, 2 Department of Midwifery, Menopause Andropause Research Center, Ahvaz Jundishapur University of Medical Sciences, Ahvaz, Iran, 3 Alimentary Tract Research Center, Clinical Sciences Research Institute, Department of Biostatistics and Epidemiology, School of Public Health, Ahvaz Jundishapur University of Medical Sciences, Ahvaz, Iran, 4 Department of Nursing, Faculty of Medical Sciences, Tarbiat Modares University, Tehran, Iran, 5 Department of Public Health and Community Medicine, Tufts University, School of Medicine, Boston, United States of America

* parvinabedi@ymail.com

## Abstract

The Labour Hopscotch Framework (LHF) is a practical tool designed for women to use with the support of their birthing partners throughout labor and birth. It was specifically developed to help midwives support women in achieving a physiological birth. We conducted a randomized controlled trial to evaluate the impact of the LHF on maternal and neonatal outcomes. The study took place from March 2023 to October 2023 in Ahvaz, Iran. A total of 124 primiparous women with full-term pregnancies were recruited and randomly allocated to either the intervention or control group. Eligible participants for the study included women who were 18 years of age or older, had cephalopelvic proportions within normal limits, were carrying low-risk singleton pregnancies, and had attended childbirth education classes. Women were excluded if they had contraindications for vaginal birth, were carrying a fetus with abnormalities, or had medical conditions that could complicate childbirth. Birth mode, average duration of the labor phases, maternal and neonatal outcomes, and overall childbirth satisfaction were assessed as outcomes.

Women who used the LHF experienced shorter labor durations compared to the control group (d = -1.3 min, 95% CI: -1.6 to -0.8). Women using the LHF were 10.3 times more likely to experience a vaginal birth (OR = 10.3, 95% CI: 1.3-88.5, p = 0.030). Women in the LHF group initiated breastfeeding earlier than those in the control group (OR = 20.4, 95% CI: 7.7-53.3, p < 0.001), but no difference between the two groups for exclusive breastfeeding (p = 0.496). Two groups did not show any significant difference in neonatal outcomes. The mean satisfaction score was higher among women with LHF compared to women in the control group (d = 2.3, 95% CI: 1.8 to 2.8), p < 0.001). The LHF can increase the rate of physiological births and reduce the duration of the labor phases. Also, women were more satisfied with their birth experiences.

**Data availability statement:** The datasets generated and analyzed during this study are not publicly available due to ethical constraints related to participant privacy. However, they can be made available from the corresponding author upon reasonable request, subject to approval by the research ethics board. Data requests must be addressed to the Reproductive Health Promotion Research Center of Ahvaz Jundishapur University of Medical Sciences, that will provide access after evaluating requests: RHPRC@ajums.ac.ir.

**Funding:** This article constituted a segment of SA's PhD dissertation. We extend our thanks to the Research Deputy of Ahvaz Jundishapur University, faculty members of the Nursing and Midwifery School, and the management and birth ward staff at Shahid Baghaei Hospital for their invaluable assistance and support. Gratitude is also extended to all study participants for their willingness to take part. Ahvaz Jundishapur University of Medical Sciences is the funder of the study (grant code: RHPRC-0125.2023-01-28). The funder provided support for the research but had no involvement in study design, data collection and analysis, decision to publish, or preparation of the manuscript.

**Competing interests:** The authors have declared that no competing interests exist.

## Trial registration

ClinicalTrials.gov IRCT20161106030750N2

## Introduction

Childbirth experience can have a considerable impact on the future health and well-being of women, children, and families [1]. According to the International Confederation of Midwives, childbirth is a unique physiological and psychological process that ideally occurs spontaneously, with minimal need for medical interventions [2]. Women's capacity to make informed decisions regarding the use of interventions during childbirth is crucial for their reproductive health and quality of care. The relationship between women and midwives is important in supporting informed decision-making [3]. Over time, childbirth practices have evolved, shifting from home-based midwifery care to predominantly hospital-based procedures [4]. This shift has led to increased medicalization of childbirth, changing its perception from a natural process to one often viewed as requiring medical intervention [5].

This trend has been associated with rising cesarean section (C/S) rates, with studies indicating that unnecessary C/S, as per World Health Organization (WHO) standards, do not improve outcomes at the population level [6,7]. Despite WHO recommendations to limit such interventions, C/S rates have continued to rise globally, with notable increases in several countries [8]. A study conducted in 169 countries showed that the average C/S rate increased from 6.7% in 1990 to 18.6% in 2014 and 21% in 2015 [9]. The cesarean rate in the United States has increased to 31.8% in the year 2020 [10]. The percentage of C/S in Iran was 53.6%, according to the most recent study conducted there in 2021[11].

According to a study in Ireland (2022), the cesarean rate was 35.4%, which is expected to continue to increase [12]. Due to the rising prevalence of C/S, the leadership midwifery management at the research site encouraged midwives to explore innovative methods to reduce interventions and promote natural, physiological childbirth. In 2015, an Irish midwife developed a visual model known as the 'Labour Hopscotch Framework (LHF) [13]. The LHF is a structured and practical tool designed to enhance the childbirth experience by promoting active participation and optimal fetal positioning during labor. This framework is intended for use by women in collaboration with their birthing partners and is specifically crafted to support midwives in facilitating a natural birth. If a birthing partner is not available, midwives can fully support the laboring women. Midwives are responsible for providing emotional support, educating on labor techniques and pain management, monitoring the progress of labor, and offering physical assistance. The primary objective of the LHF is to encourage and maintain the ideal fetal position, which is crucial for a physiological birth. This is achieved by educating both women and midwives about the importance of remaining active during labor and implementing various non-pharmacological pain management techniques. These techniques include hydrotherapy, movement, and postural adjustments, all aimed at alleviating discomfort and supporting the labor process. Each stage of the LHF is accompanied by a defined time frame and is presented in a sequential manner that aligns with the progression of labor, ensuring that interventions are timely and relevant. By integrating these practices, the LHF provides a comprehensive approach to optimizing the labor experience and promoting successful natural childbirth outcomes. LHF has seven steps (Table 6), including mobilize, stool, toilet, water, mat, birthing ball and alternative therapy, and the time allotted for each step is 20 minutes [13].

Carroll et al. in their study found that the use of the LHF resulted in less pain relief medication and a reduced frequency of C/S. The study included 809 women, with C/S rates at 8.7%,

assisted births at 14%, and spontaneous vaginal births at 77.1%. The results indicated that initiatives like LHF can help keep women active and mobile during labor, thereby promoting natural childbirth [12].

The rising rates of C/S have raised significant concerns among Iranian health policymakers and decision-makers. This highlights the urgent need for implementing effective strategies and initiatives within the healthcare system to address this issue [14]. To address the rising rate of C/S, the Iranian healthcare system is adopting several strategies. These include establishing mother-friendly hospitals, offering childbirth preparation classes for expectant mothers in public health centers for all women free of charge, implementing standardized childbirth protocols, and conducting workshops on physiological childbirth for midwives and obstetricians [15]. Despite these efforts, the rate of c/s has yet to decline [16]. Since LHF has not been examined in Iran, it is plausible to view LHF, which involves women in decision-making during labor, as a strategic intervention. This approach may improve outcomes for both mothers and neonates. The research aimed to investigate the impact of LHF on outcomes for both mothers and newborns. We hypothesize that implementing LHF could reduce the rate of unnecessary cesarean sections while simultaneously enhancing outcomes for mothers and neonates.

## Methods

### Trial design and participants

This randomized controlled trial, consisting of two parallel groups, was carried out with primiparous pregnant women at full term in Ahvaz, Iran. From March to October of 2023, this study was conducted in Ahvaz's Shahid Beqaei Hospital.

Eligible participants for the study included primiparous married women who were 18 years of age or older, had a gestational age of 37 weeks or more, planned to have a vaginal birth, had cephalopelvic proportion (In Iran, pelvic examination prior to labor is commonly used as a method to assess the fit of the fetal head with the mother's pelvis. Although the scientific evidence regarding the accuracy of this method is mixed, it remains widely utilized in many healthcare settings due to its simplicity and ease of access. To assess the cephalopelvic proportion in pregnant women, a pelvic examination was carefully conducted by an obstetrician), low-risk singleton pregnancies (a low-risk pregnancy refers to pregnancies in which the pregnant woman does not have any chronic diseases or specific medical conditions that could threaten the health of either the mother or the fetus), estimated fetal weights between 2500 and 4000 grams (according to the last ultrasound in the third trimester), basic literacy, and had attended childbirth educational classes. Women who were contraindicated for vaginal birth, had multiple fetus, had a history of abortion, were carrying an abnormal fetus, preeclampsia or eclampsia, placental abruption, placenta previa, a history of fertility problems, and medical disorders like cardiovascular, liver, renal, or brain diseases were excluded from the study.

### Setting

The research was carried out at Ahvaz's Shahid Baqaei Hospital. Ahvaz is a capital city of Khuzestan province and is one of the big cities of Iran which has ethnic and population diversity. In Iran, prenatal and maternity care are provided free of charge for all pregnant women in public health centers. Pregnant women attend eight childbirth preparation classes starting from the 20th week of pregnancy. The content of these classes includes: understanding anatomical and physiological changes during pregnancy, fetal growth and development, common pregnancy complaints, danger signs, personal and mental health, proper nutrition,

the importance of prenatal care, types of birth (natural or cesarean), the benefits and side effects of each, childbirth pain, pain reduction methods, stages of labor, various birthing positions, necessary interventions during labor, the importance of natural childbirth, the role of a birth companion, preparation of other family members, childbirth planning, stretching and relaxation exercises, proper breathing techniques, postpartum care, postpartum danger signs, family planning, and newborn care. Danger signs for the baby are also explained. The midwife uses teaching aids such as films, whiteboards, pamphlets, educational posters, and booklets, all based on valid and standard guidelines from the Iranian Ministry of Health. Although Shahid Beqaei Hospital is affiliated to the armed forces, it is a government hospital that accepts all patients. In this hospital, labour and birth are supported by midwives under the supervision of the obstetricians so that necessary measures can be taken in case of any problems.

Among those who visited the hospital clinic, pregnant women who met the eligibility criteria for the study were selected. Since one of the criteria for inclusion was a gestational age of 37 weeks, and not all women met this criterion, sampling was conducted over several months. The lead researcher (SA) provided the women with comprehensive information about the study's aims and methods, ensuring their understanding and verbal agreement to participate. Following this, the clinic secretary was responsible solely for the administrative task of distributing and collecting the written consent forms. Subsequently, each eligible participant was interviewed to complete a demographic questionnaire.

## Randomization and allocation concealment

**Generation of a random sequence.**  Randomization was conducted using the "blockrand" package in WIN PEPI software [17], with block sizes of 4 or 6. The allocation ratio between the LHF and control groups was maintained at 1:1. A total of 10 blocks of size 6 and 16 blocks of size 4 were randomly selected using WIN PEPI software. The random sequence was prepared by the statistical consultant.

**Allocation concealment.**  The research assistant, who was blinded to the study process, recorded the group allocation on a piece of paper and sealed it in opaque, sequentially numbered envelopes. These envelopes were kept by the clinic's secretary, who was also unaware of the study's objectives. Both the researchers and participants remained blinded to the group assignments until the study commenced.

**Blinding.**  Although blinding of researchers and participants was not feasible due to the nature of the study, the group assignments were concealed from the outcome assessors and data analysts.

**Intervention and follow-up.**  After assigning participants to their respective study groups, one of the researchers (SA) conducted two training sessions for the intervention group over two consecutive weeks, providing comprehensive explanations of all components of the LHF. The content of the LHF includes steps learned by the researcher (SA) during a physiological childbirth workshop and is part of the midwifery school curriculum. While the LHF was not explicitly part of the physiological childbirth workshops, the content of these workshops aligned with the key components of the LHF. The researcher (SA) used instructions approved by the framework's designer to ensure compliance with all the points compiled by the main designer of the LHF [18]. The equipment (birth balls, etc.) were already available for women to use at the study site. The TENS device was the only piece of equipment not available at the study site and was provided by the researcher. Group-based training for the Labour Hopscotch Framework (LHF) involved an average of 8–10 women per session. During the first session, the seven steps of LHF were explained to the participants to ensure they were

prepared for childbirth at 38 weeks of gestation. Additionally, it was communicated that they could have a companion during labor. In the second session, the participants' concerns and questions were addressed, and further explanations about LHF were provided if needed. Each participant was given the researcher's phone number for any questions or in case labor began. Women were requested to contact the researcher (SA) upon admission to the hospital for labor and birth. Participants were admitted during the active phase of labor (4 cm dilation), and the researcher was present as a birth attendant, implementing LHF for each participant. Furthermore, the LHF was communicated to the obstetrician overseeing their care. The researcher, acting as the primary midwife, was responsible for providing all aspects of care during the labor process, including the birth and postpartum support, thereby ensuring continuity and consistency in the participants' childbirth experience.

It is important to note that the primary responsibility rested with the woman herself, while the researcher was there to support and facilitate the birth. Labor and birth management adhered to the principles outlined in LHF. Women in the control group received only standard care according to hospital policies, with their labor and birth also managed by the researcher. A research assistant (a midwife), who was blinded to participant grouping, recorded partogram details, documented birth data, and assessed maternal and neonatal outcomes. To evaluate exclusive breastfeeding, the research assistant followed up with participants by phone six weeks postpartum. Given the unpredictability of childbirth, the researcher was available online 24/7 to ensure support for all births. If multiple women were admitted simultaneously, a trained research assistant (a midwife) managed the additional participants until the researcher was available.

## Outcomes

**Primary outcomes.** The primary outcome of this study was to examine the average duration of the first, second, and third stages of labor, as these metrics are essential for evaluating the effectiveness of the Labour Hopscotch Framework (LHF) intervention in optimizing the childbirth process.

**Secondary outcomes.** The secondary outcomes encompassed the mode of childbirth (c/s or vaginal birth), overall satisfaction with the childbirth experience, occurrence of labor augmentation (yes or no), perineal tears, Apgar scores at the first and fifth minutes after birth, neonatal intensive care unit (NICU) admissions (yes or no), initiation of breastfeeding within one-hour post-birth (yes or no), and exclusive breastfeeding at six weeks postpartum (yes or no). Pharmacological pain relief and epidurals are not used in the Ahvaz government hospitals, that's why it was not measured in the outcome.

**Sample size.** To determine the sample size, the formula for comparing the means of two groups was employed, with significance level α = 0.05 and power β = 0.1. Based on the findings of a previous studies [19, 20], the mean duration of the first stage of labor was x1 = 11.08 for the intervention group and x2 = 8.59 for the control group, with corresponding standard deviations of s1 = 4.39 and s2 = 3.35, respectively. The initial sample size was calculated to be 52 women in each group. Considering a projected dropout rate of approximately 15% throughout the study, the anticipated final sample size for each group was determined to be 62 women.

**Data collection instruments. Questionnaires assessing demographic and obstetric characteristics.** The demographic survey collected information on the woman's and her husband's age, ethnicity, education level, occupation, economic status, and body mass index (BMI). Additionally, it documented various obstetric features, including the history of prenatal care and attendance at childbirth preparation classes. In our study, measuring the age and employment status of husbands was essential due to the cultural and socio-economic context in Iran, where

the role of the husband/partner is often strongly tied to family decision-making, particularly in areas such as healthcare access, financial support, and childbirth decisions.

## Partogram form

Partograms are used by midwives and obstetricians as a valuable instrument for recording the details of labour and are recognized in both developed and developing countries [21]. The partogram in our study recorded information on the length of labor phases, results of vaginal examinations, and the status of the perineum.

## Checklist for maternal and neonatal outcomes

A checklist was used for recording maternal and neonatal outcomes included details on the mode of childbirth, reasons for cesarean sections, NICU admissions, Apgar scores at 1 and 5 minutes after birth, initiation of breastfeeding within the first hour, exclusive breastfeeding 6 weeks postpartum, and measurements of the neonate's weight, head circumference, and length. The validity of the demographic questionnaire and the checklist was evaluated through face and content validity. Face validity was established with input from midwifery and reproductive health specialists, who reviewed the tool items for difficulty, relevance, and clarity, with adjustments made based on their feedback. Content validity was ensured through feedback from 10 specialists, focusing on grammar, precise wording, proper word order, and appropriate scoring [22, 23].

## Mackey's scale for assessing childbirth satisfaction

This scale contains 34 questions to gauge women's satisfaction and experience with birth. Based on cultural issues, items 12 and 13 were removed from the scale during psychometric evaluation of the scale in Iran [24]. MacKay's childbirth satisfaction questions rated on a 5-point Likert scale (extremely unsatisfied to highly satisfy) from 1 to 5 points and the total score varied from 32 to 160. A score of 128 and above was considered as a good satisfaction. In this questionnaire, satisfaction was examined in 5 dimensions: self-satisfaction, satisfaction with spouse, satisfaction with baby, satisfaction with midwife, overall satisfaction, and satisfaction with doctor. Scoring for questions 33 to 36 involved a four-point Likert scale, where responses ranged from (1): very negative to (4): very positive. A cumulative score of ≥ 12 indicates positive experiences and scores below 12 indicate negative experiences [23].

In 2003, Goodman assessed the reliability and validity of Mackey's childbirth satisfaction rating scale [25]. In Iran, Moudi et al. conducted the psychometric evaluation of the questionnaire, confirming its reliability with a Cronbach's alpha of 0.78 [24] which shows that this questionnaire is reliable.

Two groups completed the Mackey's Childbirth Satisfaction Rating Scale 24 hours after birth and prior to hospital discharge.

## Data analysis

The analysis of data utilized SPSS version 22 software [26]. Categorical variables were summarized as counts and percentages, and continuous variables were summarized using means, standard deviation and ranges. Effect size (Cohen's d), calculated as the difference in means divided by the pooled standard deviation with 95% confidence intervals, was used to assess the magnitude of differences between the intervention and control groups. The analysis was done per protocol. Unconditional logistic regression was performed to control potential confounders of binary outcomes.

## Ethical consideration

The Research Ethics Committee of Ahvaz Jundishapur University of Medical Sciences granted approval for the study protocol and informed consent documents (IR.AJUMS.REC.1401.512) on 20 January 2023. The authors affirm that all ongoing and related trials for this intervention have been registered. Additionally, the study was submitted to the Iranian Clinical Trial Center (IRCT) with registration number IRCT20161106030750N2 on 20 February 2023. Written informed consent was obtained from all participants prior to their involvement in the study. The research adhered to the approved guidelines and followed the CONSORT 2010 Statement (Consolidated Standards of Reporting Trials) [27].

## Results

### Participants

Participating were 124 eligible women that were recruited between March and October 2023, who were randomized into two groups: LHF (n = 62) and control (n = 62) (Fig 1). Two participants, one from each group, withdrew from the research. The reason for withdrawal in the intervention group was a lack of response from the participant, while in the control group, the participant decided to undergo a C/S at another hospital.

Table 1 provides a comprehensive overview of the baseline characteristics of the participants. Women in the intervention group had an average age of 22.4 (SD 4.5) years, slightly lower than the control group's average of 22.6 (SD 4.8) years. Most women in both the intervention and control groups were homemakers, comprising 96.8% and 93.6%, respectively. The majority of participants (59.7% and 50%, respectively) in the intervention and control groups completed high school and obtained a diploma. In both groups, the average gestational age at the first prenatal care session was 8 weeks and 2 days, as indicated in Table 1. In terms of economic status, age of husband, employment status of husband, education level of husband, and BMI, there was no substantial difference detected between the two groups.

### Maternal outcomes

Table 2 presents the maternal outcomes of the participants in this study. Two groups did not differ in terms of gestational age at birth. Notably, women who used the LHF demonstrated markedly higher rates of vaginal birth in contrast to the control group (98.4% vs. 85.2%).

Women who used the LHF experienced shorter first, second, and third stages of labor and shorter total labor duration in contrast to the control group (mean 162.6, SE 55 min vs. mean 260.30, SE 95.3 min). For the total labor duration, a large negative effect size was observed (d = -1.3, 95% CI: -1.6 to -0.8), indicating a substantial difference favoring the intervention group. The first stage of labor also demonstrated a significant negative effect size (d = -1.2, 95% CI: -1.6 to -0.8). Similarly, the second stage of labor revealed a large negative effect size (d = -0.9, 95% CI: -1.3 to -0.5). Significant differences were also noted for the third stage of labor (d = -0.8, 95% CI: -1.2 to -0.4). In this study, negative effect size indicates that participants in the intervention group had shorter labor durations compared to those in the control group. Specifically, the total duration of labor was significantly reduced in the intervention group. Likewise, the first, second, and third stages of labor were all shortened. These negative effect size highlight the substantial reduction in labor duration, favoring the intervention group. These results suggest that the LHF intervention facilitated a more efficient labor process, potentially leading to improved maternal and neonatal outcomes.

Participants in the LHF group had a decreased likelihood of undergoing episiotomy (66.7% vs. 90.4%) (OR = 1.6, 95% CI: 0.2–14.3) and receiving oxytocin for augmentation in

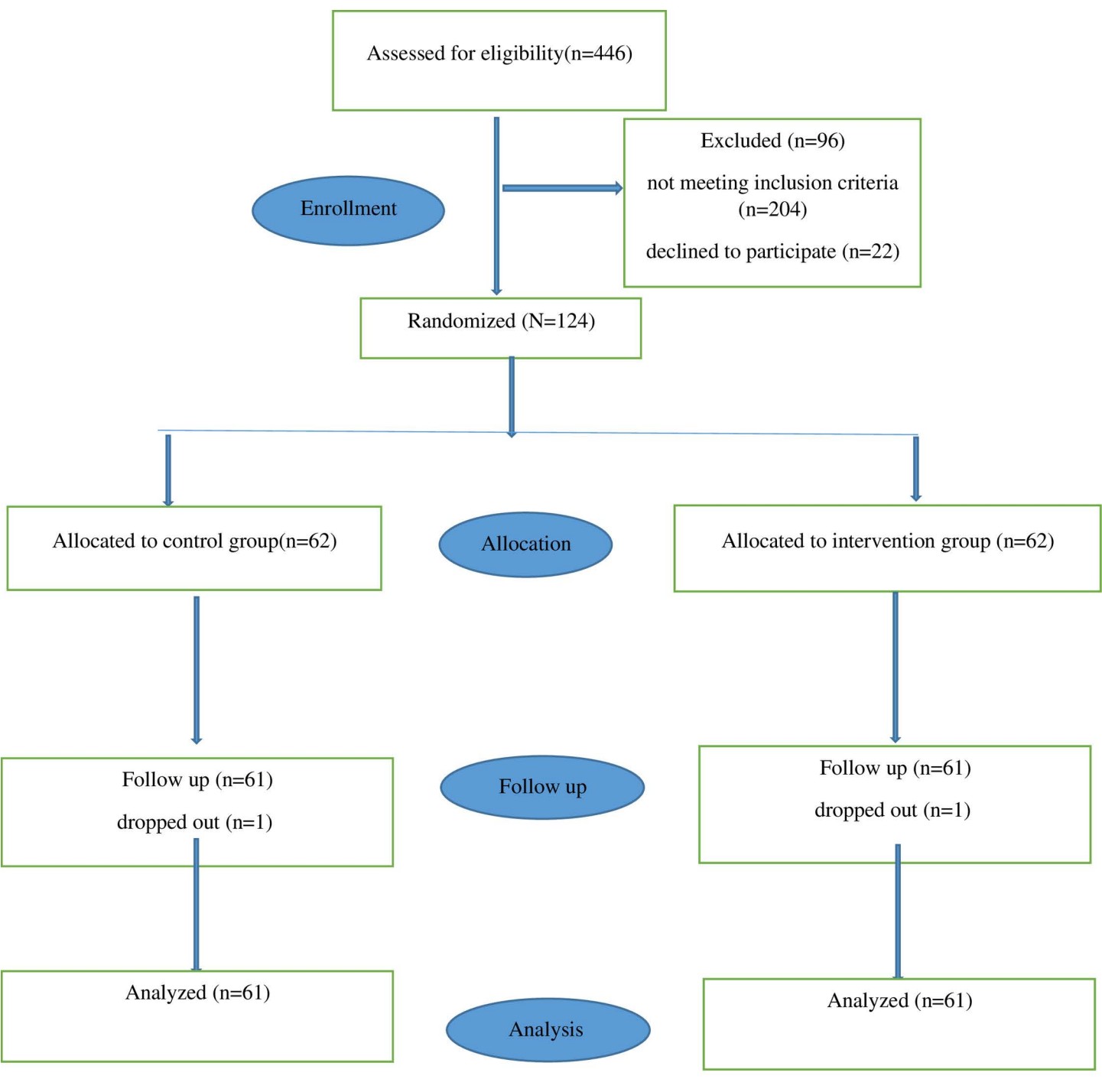

Figure1: Recruitment and retention process flowchart for study participants

**Fig 1. Recruitment and retention process flowchart for study participants.**

**Table 1. Demographic and obstetric features of intervention and control groups at the baseline.**

| Characteristics | Intervention group (n = 62) | Control group (n = 62) |
|---|---|---|
| | Mean ± SD | |
| Age (y) | 22.4 ± 4.5 | 22.6 ± 4.8 |
| BMI (kg/m²) | 23.8 ± 3.4 | 23.8 ± 4.3 |
| Education level | | |
| | N(%) | |
| Primary and secondary school | 12 (19.3) | 16 (25.8) |
| High school Diploma | 37 (59.7) | 31 (50) |
| University degree | 13 (21) | 15 (24.2) |
| Employment status | | |
| Housewife | 60 (96.8) | 58 (93.5) |
| Employed | 2 (3.2) | 4 (6.5) |
| Economic status | | |
| Poor | 17 (27.4) | 19 (30.6) |
| Moderate | 32 (51.6) | 28 (45.2) |
| Good | 13 (21) | 15 (24.2) |
| First prenatal care (weeks) | 8.7 ± 1.9 | 8.6 ± 2 |

Values are expressed as no (%) or mean ± standard deviation; unless otherwise stated.

comparison to the control group (11.5% vs. 57.6%) (OR = 10, 95% CI: 0.1–0.2). The logistic regression model indicated that participants in the LHF group were over 10 times more likely to achieve a vaginal birth (Table 3). (OR = 10.3, 95% CI: 1.3–84.5).

## Neonatal outcomes

Table 4 displays the neonatal outcomes of the participants. With respect to the APGAR scores of the newborns at one and five minutes (p = 0.129 and p = 0.100, respectively), no significant differences were observed between the two groups. After birth, women in the LHF group initiated breastfeeding sooner than the control group (OR = 20.4, 95% CI:7.7–53.3). Nevertheless, there was no difference between the two groups in terms of exclusive breastfeeding in six weeks postpartum.

## Childbirth satisfaction and positive experience

Women in the LHF group exhibited greater mean satisfaction (mean 142.1, SE 0.6) compared to those in the control group (mean 116.8, SE 2) (p < 0.001). The LHF group consistently scored higher across all categories of the satisfaction scale, except for satisfaction with the physician (p = 0.289) (Table 5). The results indicated significant differences in satisfaction measures between the intervention and control groups. For the total satisfaction score, a large effect size was observed (d = 2.3, 95% CI: 1.8 to 2.8), indicating a substantial difference between the two groups. The self-satisfaction score also showed a large effect size (d = 3.3, 95% CI: 2.7 to 3.9). Satisfaction with the midwife revealed a similarly large effect size (d = 2.3, 95% CI: 1.8 to 2.7).

However, for satisfaction with the physician, the effect size was smaller and non-significant (d = 0.2, 95% CI: -0.1 to 0.6). Significant differences were observed for satisfaction with the baby (d = 1, 95% CI: 0.6 to 1.4), overall childbirth satisfaction

**Table 2. Maternal outcomes in the intervention and control groups.**

| Characteristics | Intervention group (n = 61) | Control group (n = 61) |
|---|---|---|
|  | n (%) or Mean ± SE |  |
| Gestational age at birth (wk) | 39.5 ± 0.1 | 39.5 ± 0.1 |
| Birth mode |  |  |
| Normal vaginal birth | 60 (98.4) | 52 (85.2) |
| C/S | 1 (1.6) | 9 (14.8) |
| Amniotic membrane |  |  |
| Intact | 45 (73.8) | 30 (50) |
| Ruptured | 13 (21.3) | 28 (46.7) |
| Meconium | 3 (4.9) | 2 (3.3) |
| Augmentation of labor | 7 (11.5) | 34 (57.6) |
| Perineal status |  |  |
| Episiotomy | 40 (66.7) | 47 (90.4) |
| 1st degree laceration | 18 (30) | 1 (1.9) |
| 2nd degree laceration | 0 (0) | 2 (3.8) |
| Intact | 2 (3.3) | 2 (3.8) |
| Length of labor, min |  |  |
| Stage 1 | 124.5 ± 6.1 | 200.1 ± 10.6 |
| Stage 2 | 32.9 ± 1.5 | 55.6 ± 4.5 |
| Stage 3 | 5.2 ± 0.2 | 7.5 ± 0.5 |
| Total duration | 162.6 ± 7.1 | 260.4 ± 12.1 |
| Causes of c/s |  |  |
| Fetal distress | 1 (100) | 3 (33.3) |
| Meconium | 0 | 2 (22.2) |
| Labor stop | 0 | 1 (11.1) |
| No response of Augmentation | 0 | 2 (22.2) |
| Elective | 0 | 1 (11.1) |

Values are expressed as no (%) or mean (standard errors); unless otherwise stated.

(d = 2.3, 95% CI: 1.9 to 2.8), and positive experience score (d = 2.7, 95% CI: 2.2 to 3.2), all indicating large effect sizes.

## Satisfaction with LHF

All participants in the intervention group reported contentment with the use of the LHF, and 43.3% of women considered all steps of LHF useful. They were more pleased with water step than the other steps (Table 6).

## Discussion

This study evaluated the impact of the LHF on both maternal and neonatal outcomes among pregnant women. The results indicated that the LHF reduced the need for unnecessary medical interventions, such as episiotomy and labor augmentation, shortened the duration of labor stages, and increased the likelihood of a physiological birth. Additionally, the LHF contributed to improved childbirth experiences and greater satisfaction among mothers. However, no significant differences were observed between the two groups concerning NICU admissions

**Table 3. Unadjusted and adjusted odds ratios for factors associated with normal vaginal birth.**

| Variables | Intervention group (n = 61) | Control group (n = 61) | Unadjusted analysis | p-value[**] | Adjusted analysis | p- value[***] |
|---|---|---|---|---|---|---|
| | | | Odds ratio (95% CI) | | Odds ratio (95% CI) | |
| LFH[a] | 0% | 100% | 10.4 (1.27–84.73) | 0.029 | 10.3 (1.26–84.53) | 0.030[b] |
| Education level | | | 0.788 (0.3–2.1) | 0.627 | – | |
| Primary and secondary school | 12(19.3%) | 16(25.8%) | | | | |
| High school Diploma | 37(59.7%) | 31(50%) | | | | |
| University degree | 13(21%) | 15(24.2%) | | | | |
| Socioeconomic status | | | 0.54 (0.21–1.38) | 0.196 | – | |
| poor | 17(27.4%) | 19(30.6%) | | | | |
| moderate | 32(51.6%) | 28(45.2%) | | | | |
| good | 13(21%) | 15(24.2%) | | | | |
| BMI | | | 1.12 (0.92–1.36) | 0.247 | – | |
| First prenatal care | | | 1.2(0.83–1.72) | 0.339 | – | |

[a]Labour Hopscotch Framework

[**]Back ward method

[***]Adjusted for LFH

or APGAR scores at one and five minutes after birth. While LHF improved the initiation of breastfeeding within the first hour, there were no significant difference in the rates of exclusive breastfeeding between the two groups at six weeks postpartum.

Given that the LHF was implemented for the first time in Iran, the results indicated that this framework was associated with an increase in the normal vaginal birth rate within the study sample. The findings of this study suggest that training women and involving them actively in the birthing process can enhance the rate of normal births. However, the literature remains inconclusive regarding the relationship between birth preparation, women's participation in the birth process, and the mode of birth. Our results are consistent with those of Mohaghegh et al.,

**Table 4. Neonatal characteristics, and outcomes in the intervention and control groups.**

| Variables | Intervention group (n = 61) | Control group (n = 61) |
|---|---|---|
| | n (%) or mean ± SE | |
| Neonate's sex | | |
| Male | 33 (54.1) | 28 (45.9) |
| Female | 28 (45.9) | 33 (54.1) |
| Birth weight (g) | 3227 ± 50.1 | 3304.1 ± 48.1 |
| Length (cm) | 49 ± 0.2 | 49.1 ± 0.2 |
| Head circumference (cm) | 34.1 ± 0.1 | 34.4 ± 0.1 |
| 1st min. Apgar | 9 | 8.9 ± 0.1 |
| 5th min. Apgar | 10 | 9.9 |
| Initiating breastfeeding within 1 h after birth | 51 (83.6) | 14 (23) |
| Exclusive breastfeeding 6 weeks after birth | 61 (100) | 59 (98.3) |
| NICU admission | 0 (0) | 3 (4.9) |

Values are expressed as no (%) or mean (standard errors); unless otherwise stated.

**Table 5. Positive experience and childbirth satisfaction in intervention and control groups.**

| Variables | Intervention group (n = 60) | Control group (n = 56) | p-value | Effect Size (Cohen's d) | 95% Confidence Interval |
|---|---|---|---|---|---|
| | Mean ± SE | | | | |
| Total satisfaction score | 142.1 ± 0.6 | 116.9 ± 2 | <0.001[a] | 2.3 | (1.8–2.8) |
| Self-satisfaction score | 44.2 ± 0.2 | 31.1 ± 0.7 | <0.001[a] | 3.3 | (2.7–3.9) |
| Satisfaction with midwife | 44.9 ± 0.1 | 37.9 ± 0.6 | <0.001[a] | 2.3 | (1.8–2.7) |
| Satisfaction with physician | 24 ± 0.5 | 23 ± 0.8 | 0.289[a] | .2 | (-.1-.6) |
| Satisfaction with baby | 15 ± 0.2 | 13.6 ± 0.3 | <0.001[a] | 1 | (.6–1.4) |
| Overall childbirth satisfaction | 14.2 ± 0.8 | 11.1 ± 0.2 | <0.001[a] | 2.3 | (1.9–2.8) |
| Positive experience score | 13.8 ± 0.1 | 9.7 ± 0.2 | <0.001[a] | 2.7 | (2.2–3.2) |

Values are expressed as mean (standard errors); unless otherwise stated.

[a]Independent t-test.

who found that women with a strong preference for vaginal birth had significantly higher rates of successful vaginal deliveries [23]. Our study's findings are consistent with those of Carroll et al., suggesting that the LHF can facilitate physiological births and increase the chances of normal vaginal deliveries [12]. However, other research has found that some interventions and educational efforts during pregnancy may not affect the mode of birth or could even lead to higher rates of cesarean sections, which contrasts with our results [28, 29]. This variation could be due to differences in the populations studied or the nature of the interventions implemented.

The present study indicates a correlation between using the LHF and a lower incidence of oxytocin utilization for augmentation, similar to the findings of Carroll et al. [12] and Mohaghegh et al. [23]. A systematic review and meta-analysis indicated that prenatal classes reduce anxiety, which leads to a decrease in adrenaline levels, and also increases the level of endogenous oxytocin, which leads to efficient uterine contractions [30].

**Table 6. Satisfaction with Labour Hopscotch Framework.**

| Most Beneficial Steps | N(%) |
|---|---|
| Mobilize | 1 (1.7) |
| Stool | 2 (3.3) |
| Water | 2 (3.3) |
| Mobilize & water | 5 (8.3) |
| Water & toilet | 1 (1.7) |
| Water & stool | 2 (3.3) |
| Water & mat | 2 (3.3) |
| Water & birthing ball | 10 (16.7) |
| Stool & birthing ball | 4 (6.7) |
| Stool & mobilize | 5 (8.3) |
| All steps | 26 (43.3) |
| Not satisfied | 0 |

We found that the rate of episiotomy and 2nd and 3rd degree tears in the perineum was reduced in the group that used LHF in contrast to the control group, and the rate of 1st degree tears was higher in the LHF group. Our results align with a systematic review indicating that maintaining a flexible sacral position during labor results in reduced perineal trauma [31]. These findings explain that the positions used in LHF during labor can reduce trauma to the perineum.

Our findings indicated a reduction in the duration of various stages of labor in the group that employed the LHF. These results align with a systematic review, which demonstrated that maternal birth positions during labor can significantly shorten labor stages [32]. Similarly, research by Mohaghegh et al. observed that women who incorporated upright birthing positions, such as squatting, using a birthing ball, and walking, experienced a decrease in the length of labor phases [23]. A plausible explanation for these findings is that upright positions, the use of a birthing ball, and walking facilitate better fetal head rotation, thereby accelerating labor progression [32]. However, Zang et al. reported that the flexible positioning of the sacrum did not affect the duration of labor stages in their study. They attributed this contradiction to differences in the parity of the participating women and the onset timing of the second stage of labor [31].

In the present study, women participated in the intervention group reported high level of satisfaction with using LHF. Pregnancy and childbirth are two very important and extraordinary moments in a woman's life. Providing support to pregnant women and facilitating their participation in the labor process can enhance their satisfaction with childbirth [23]. Effective support and engagement should be initiated at the time of admission and continually improve over the course of the birthing process. Contentment with childbirth signifies the mother's positive sentiments toward the experience, conveying a sense of active involvement, fulfillment of needs and expectations, control, inner strength, self-confidence, self-care, and support [33]. Women used the LHF reporting greater satisfaction and a more positive labor and birth experience, neither the control nor intervention groups expressed high levels of satisfaction with the doctor. A possible explanation for this finding is the care provided by midwives during labour and birth, and the physician was less involved. Our findings align with the research conducted by Mirghafourvand et al., which identified that a positive and supportive relationship with the midwife, maternal involvement in decision-making, and the mother's freedom of action are critical components influencing childbirth satisfaction [34]. Freedom of action during childbirth refers to a mother's ability to make her own decisions, such as choosing labor positions, managing pain, and accepting or refusing medical interventions. This autonomy not only empowers the mother but also enhances her satisfaction with the childbirth experience [35]. However, Afshar et al. found that support, good communication with midwives, and participation in decision-making and choice were not significant factors in women's satisfaction with childbirth. A possible explanation for this finding is that Afshar et al. selected participants from women with higher socioeconomic status (SES), which may have influenced their perceptions and expectations regarding childbirth [28]. Research indicates that women with higher socioeconomic status (SES) often have greater access to resources and decision-making power compared to those with lower SES [36]. Consequently, it is reasonable to anticipate that the LHF may be particularly beneficial in empowering women from lower socioeconomic backgrounds, while also providing valuable support to women from diverse socioeconomic statuses. Carroll et al. found that women who used the LHF considered it beneficial and expressed a preference for learning about it earlier in their pregnancies, suggesting that early introduction of the framework could enhance its effectiveness in subsequent births [37].

Numerous studies have indicated that the duration of labor stages significantly influences neonatal outcomes, with prolonged labor associated with heightened risks of complications for the newborn [38]. From the findings of the present study, we concluded that women using the LHF had a higher probability of starting breastfeeding earlier after birth. In the explanation of these findings, it can be said that the shorter the duration of labor, the faster the initiation of breastfeeding [23]. The results of this study indicate that there were no distinctions observed in the 1st and 5th minute Apgar scores between the LHF and control groups. Our findings are in agreement with those of previous studies, showing that the LHF did not affect Apgar scores, and there was no need for NICU admissions for the neonates [19,23].

Our findings indicated that the components of the LHF considered most beneficial by women included all steps, with particular emphasis on water, the birthing ball, and mobilization. This aligns with the results of Carroll et al., who found that women using LHF expressed high satisfaction with the program, noting that it is both easy to implement and effective. This may explain why women are receptive to the program [12].

This research was conducted in a medical facility with a high rate of cesarean sections. The study's significant findings highlight the need for education and support to help women make informed decisions about their birth choices. Additionally, all hospitals in Iran should provide the necessary facilities for physiological childbirth to enhance the childbirth experience and reduce fear. To lower the cesarean rate in Iran, policymakers should prioritize the expansion of midwifery services, particularly those supporting physiological childbirth. By empowering midwives to facilitate natural labor processes and providing comprehensive care, the likelihood of unnecessary cesarean sections can be reduced.

## Study's strengths and limitations

This study represents the first randomized controlled trial conducted in Iran to evaluate the impact of the LHF on maternal and neonatal outcomes. While this study has several strengths, there are also two key limitations. First, both the researchers and participants were not blinded, which may have introduced bias. Second, the study was conducted in a hospital affiliated with the military. Although this hospital serves women outside the military community, it differs from educational hospitals in that medical residents were not involved in providing care during labor and birth. Therefore, the generalizability of our results to other settings, such as educational hospitals, may be limited.

## Conclusion

The Hopscotch Labor Framework increases the rate of normal vaginal birth, increase the initiation of breastfeeding, and women's satisfaction with childbirth. Using this method could significantly reduce the number of episiotomies, augmentation of labor and duration of different stages of labor. Based on the obtained results, the Hopscotch Labor Framework can be taught to midwives, midwifery students, nurses, and physicians for use in childbirth preparation classes.

## Supporting information

**S1 File. English protocol.**
(DOCX)

**S2 File. Persian protocol.**
(DOCX)

**S3 File. CONSORT Checklist.**
(DOC)

## Acknowledgments

Gratitude is extended to all study participants for their willingness to take part.

## Author contributions

**Investigation:** Saeedeh Askari, Mina Iravani, Parvin Abedi, Eesa Mohammadi, Shayesteh Jahanfar.

**Methodology:** Saeedeh Askari, Parvin Abedi, Bahman Cheraghian.

**Project administration:** Parvin Abedi.

**Software:** Bahman Cheraghian.

**Supervision:** Parvin Abedi.

**Validation:** Saeedeh Askari, Bahman Cheraghian.

**Writing – original draft:** Saeedeh Askari, Mina Iravani, Parvin Abedi, Eesa Mohammadi, Shayesteh Jahanfar.

**Writing – review & editing:** Saeedeh Askari, Mina Iravani, Parvin Abedi, Eesa Mohammadi, Shayesteh Jahanfar.

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
