## [Decision Letter · Decision Letter 0]

10 Sep 2024

PONE-D-24-31506The effect of Labour Hopscotch Framework on maternal and neonatal outcomes in pregnant women: A randomized controlled trialPLOS ONE

Dear Dr. Abedi,

Thank you for submitting your manuscript to PLOS ONE. After careful consideration, we feel that it has merit but does not fully meet PLOS ONE’s publication criteria as it currently stands. Therefore, we invite you to submit a revised version of the manuscript that addresses the points raised during the review process.

**ACADEMIC EDITOR: Please respond to all reviewers comments**==============================

We look forward to receiving your revised manuscript.

Kind regards,

Ahmed Mohamed Maged, MD

Academic Editor

PLOS ONE

Journal Requirements:

"This article constituted a segment of SA's PhD dissertation. We extend our thanks to the Research Deputy of Ahvaz Jundishapur University, faculty members of the Nursing and Midwifery School, and the management and birth ward staff at Shahid Baghaei Hospital for their invaluable assistance and support. Gratitude is also extended to all study participants for their willingness to take part. Ahvaz Jundishapur University of Medical Sciences is the founder of the study (grant code: RHPRC-0125.2023-01-28)."

"This article constituted a segment of SA's PhD dissertation. We extend our thanks to the Research Deputy of Ahvaz Jundishapur University, faculty members of the Nursing and Midwifery School, and the management and birth ward staff at Shahid Baghaei Hospital for their invaluable assistance and support. Gratitude is also extended to all study participants for their willingness to take part. Ahvaz Jundishapur University of Medical Sciences is the founder of the study (grant code: RHPRC-0125.2023-01-28)."

"This article constituted a segment of SA's PhD dissertation. We extend our thanks to the Research Deputy of Ahvaz Jundishapur University, faculty members of the Nursing and Midwifery School, and the management and birth ward staff at Shahid Baghaei Hospital for their invaluable assistance and support. Gratitude is also extended to all study participants for their willingness to take part. Ahvaz Jundishapur University of Medical Sciences is the founder of the study (grant code: RHPRC-0125.2023-01-28)."

4. In the online submission form, you indicated that [The datasets generated and analyzed during the current study were available from the corresponding author on reasonable request.]. 

Reviewers' comments:

Reviewer's Responses to Questions

**Comments to the Author**

1. Is the manuscript technically sound, and do the data support the conclusions?

Reviewer #1: Yes

Reviewer #2: Yes

Reviewer #3: No

2. Has the statistical analysis been performed appropriately and rigorously? 

Reviewer #1: Yes

Reviewer #2: No

Reviewer #3: I Don't Know

3. Have the authors made all data underlying the findings in their manuscript fully available?

Reviewer #1: Yes

Reviewer #2: No

Reviewer #3: Yes

4. Is the manuscript presented in an intelligible fashion and written in standard English?

Reviewer #1: No

Reviewer #2: Yes

Reviewer #3: Yes

5. Review Comments to the Author

Reviewer #1: Thank you for the opportunity to review your paper regarding the use of the Labour Hopscotch Framework in primiparous women in Iran. This is an important study as it demonstrates the use of the Framework outside of Ireland, in a completely different maternity care context and with good outcomes.

Along with the suggestions below, I encourage to you fix the typographical errors and consider having the paper professionally edited as there are many sentences that don’t quite make sense to a native English speaker. I have corrected some of the English below, however you will need to compare my version with yours to see where the changes have been made as it is difficult to differentiate in the review system. I look forward to reading the next revision.

Abstract

Line 22 - We evaluated the impact of the LHF on maternal and neonatal outcomes. Between March 2023 and 22 October 2023, a randomized controlled trial including two parallel groups was carried out with 124 primiparous women with a term pregnancy (37 weeks or longer) in Ahvaz, Iran.

Line 24 – Women who were 18 years of age without cephalopelvic disproportion, with a low-risk singleton pregnancy, and who attended specialised childbirth education classes were among the eligible participants for the study.

Line 26 - Women who were contraindicated for vaginal birth, were carrying an abnormal fetus, or had one or more medical disorders were excluded from the study.

Line 30 - Women in the LHF group initiated breastfeeding earlier than the control group after birth

Line 33 - The mean satisfaction score was higher among women using the LHF

Introduction

Line 48 – substantialrole (needs a space between words)

Line 49 – ‘The care provided to women during childbirth has undergone significant changes during these years.’ What years are the authors referring to here?

Line 92 - Since there hasn't been any examination of LHF in Iran, it is possible to see LHF assists women in decision-making during labour as a strategic intervention

Could the authors please provide a better description of the LHF and consider using a figure to illustrate its use. Based on the description here I had difficulty understanding what it was until I looked up the original Carroll et al paper, which gave me a better idea of what the LHF is and how it is used. I don’t get this from your paper.

Information about the Iran maternity system would also provide some context for your readers to help them better understand how women receive maternity care in Iran. For example, do women have choice in how they access maternity care, and are there different models of care available? If so, these could all impact how the LHF was received.

Methods

Line 104 – Please explain why being married was part of the inclusion criteria.

Line 105 – how was cephalopelvic disproportion assessed?

Line 104 – Please correct the English in this sentence. ‘Primiparous married women who were 18 years of age or older, gestational ages of 37 weeks or longer, plans for a vaginal birth, cephalopelvic proportion, low-risk singleton pregnancies, estimated fetal weights between 2500 and 4000 grams (According to the last ultrasound in the third trimester), basic literacy, and attendance at childbirth education classes were among the eligible participants for the study.’ Also, you should state that the childbirth classes were specific to the LHF.

Line 105 – ‘Although Shahid Baghai hospital is affiliated to the armed forces, it accepts all people.’ Is this hospital therefore a government run hospital? Please clarify as you later refer to government hospitals not offering epidurals or pharmacological pain relief. This will add context.

Line 122 – it is a risk to have a lay person obtaining informed consent. Please explain why the clinic secretary undertook this task.

Line 145 – was the physiological childbirth workshop specific to the LHF? Please clarify.

Line 149 – why was the TENS device excluded?

It is unclear if the researcher was the primary midwife supporting the woman in labour or if she was there in a support role. Please clarify.

Line 193 – please clarify why husband’s age was important demographic information to collect

Line 227 – please provide comment on the Cronbach’s alpha, whether this is considered reliable.

Line 237 – ‘Due to the fact that one person from each group had a drop’. Please explain what a drop is? Or do you mean one person from each group dropped out?

Results

Lines 263 – 265 – Measuring age and employment status of husbands/partners is something that is rarely done in English-speaking countries unless the study is specific to fathers so it would be good to provide some context around why this was important to include in your study. These potentially are cultural nuances that might not be understood outside of Iran.

Discussion

Line 303 – You can’t say that the LHF has increased the normal vaginal birthrate in Iran. All you can say is that using the LHF increased the NVB rate in the study sample.

Line 339 – ‘Supporting pregnant women and making them participate in the labor process.’ I think you can facilitate women to participate in the labour process but not make them. Please reconsider this sentence.

Line 344 – ‘Despite women with LHF.’ Women used the LHF, they didn’t have it – please rephrase.

Line 346 – Do you mean most of the care was provided by midwives?

Line 350 – What do you mean by ‘freedom of action’? Please clarify.

Lines 351-354 Please clarify these two sentences, beginning with Nevertheless… etc. They don’t make sense in their current format.

Line 368 – the steps of the LHF need to be included in the introduction so that the reader understands what they are, and you could also refer them to Table 6.

Line 384 – ‘was affiliated with the army.’

Conclusion

Line 389 – episiotomy should be plural in this sentence ie. episiotomies

Table 2

Please amend ‘rapture’ to ruptured’ in the table (in reference to the amniotic membrane)

Reviewer #2: This is a important piece of research. I have a few comments to help improve the analysis and reporting.

Abstract

- in line 30-31 where you say that women in the LHF group started breastfeeding earlier than the control group and in line 32 where you say there was no difference in exclusive breastfeeding at six weeks, you should include an estimate of effect with 95% confidence alongside the p-value

Introduction

- missing 'the' in line 57; should say "...the world."

Methods

- title on line 100 could be simplified to 'trial design and participants'

- the number of participants is usually reported in the sample size calculation, not in the trial design section

- the outcomes should be described better. Generally, most studies have one primary outcome (the one linked to the main objective of the trial and the one upon which the sample size is calculated); therefore it was unusual to see multiple primary outcomes cited. Additionally, each outcome should be described more clearly; for example how was mode of birth measured? Was it a binary or continuous outcome? How was satisfaction with childbirth measured? etc.

- the sample size calculation is not clearly described. The authors indicate that this was a sample size calculation to compare two ratios - do you mean proportions? Which outcome is being considered in this calculation? The authors say something about the average length of stages of labour, but this is not a ratio or a proportion, is it?

- there is no role for tests to assess normality of variables in an RCT. There is also no role for chi-squared tests, t-tests, Mann Whitney tests or ANCOVA to compare participant characteristics - this should simply be a descriptive analysis.

- for analyses comparing the primary and secondary outcomes between the arms, you should generally use regression methods to estimate effects with 95% confidence intervals and p-values, unless there are good reasons to do so, e.g. a continuous outcome that is a-priori known to not be normally distributed therefore typically analysed using non-parametric methods.

Results

- given my comment regarding the statistical methods, the analysis in Tables 1, 2, 4 and 5 should be purely descriptive; there should be no statistical tests comparing the groups

- Table 3 should include columns showing the proportions and denominators of the outcome in each arm (see Table 4 in https://www.thelancet.com/journals/lancet/article/PIIS0140-6736(18)31782-3/fulltext for an example of what I mean.)

- Wherever results pertaining to the primary and secondary outcomes are cited in the text, you should include effect estimates with 95% confidence intervals and p-values where these have been estimated; citing group proportions and p-values alone e.g. on line 269 is not sufficient.

Overall I would recommend consulting a statistician with experience in analysing and reporting clinical trials to improve the statistical methods and results.

Reviewer #3: Thank you for the opportunity to review this manuscript, entitled “The effect of Labour Hopscotch Framework on maternal and neonatal outcomes in pregnant women.” This article presents a randomized controlled trial exploring an association between maternal and neonatal outcomes with use of the Labour Hopscotch Framework (LHF), a visual birthing tool to facilitate movement during labor and physiologic birth. While I commend the authors on undertaking research to reduce unnecessary cesarean births and optimize health outcomes, this research has extremely limited generalizability, making it of questionable value to the journal’s international readership. In addition, there are notable concerns of a biased sample and lack of a health equity lens. Areas for major improvement are outlined below. In light of the significant concerns raised, I would respectfully suggest that the authors reconsider reframing this article as a quality improvement project rather than an original research article to increase its utility to other care providers and facilities.

Poorly described inclusion/exclusion criteria

• Inclusion criteria included “cephalopelvic proportion, low-risk singleton pregnancies,” however, the authors do not describe how these characteristics were determined or defined. Routine screening for cephalopelvic proportion antenatally is of very questionable clinical utility and, depending on the assessment tool used, involves unnecessary exposure to radiation or invasive examination.

• Attendance at childbirth education classes was required for participation, but the content, format, and timing of classes was not described.

• The authors state that the use of the LHF requires the active assistance of “birthing partner,” however they do not list the presence of this partner among the inclusion criteria.

• The authors do not describe the number of eligible participants who declined study participation.

Concerns regarding health equity

• The rationale for requiring attendance at childbirth education classes for participation is not justified and likely prevented persons of lower socioeconomic status from participating in the study.

• The requirement for active participation of the “birthing partner” raises concerns about the value of this research from a health equity lens for persons who did not present with a partner or whose partner was not willing or able to fill this role. No mention or providing a partner for those without one is described.

• Why were women with a history of abortion excluded?

• The authors make statements that show bias based on SES such as “Women with higher SES have more choice and control over their life than women with lower SES.”

Limited generalizability

• The same researcher was present at all labor/births in the study. The authors do not describe the researcher’s involvement or the ways in which their presence may affected the care provided or implementation of the intervention.

• No pharmacological pain management options, including epidural analgesia, were available at the research site. This is not the norm in most birth settings.

• The sample is not representative of the general childbearing population. All participants were the wives of military persons, with 90-96% being unemployed “housewives.” Median age, BMI, and onset of prenatal care are well below normal ranges for childbearing persons in most settings. These limitations are not addressed.

• A 1.6% cesarean section rate in the intervention group (n=1) is well below international norms and best practice standards. Implications are not sufficiently explored.

• The details of LHF implementation are not sufficiently described, making it difficult to consider how this intervention might be implemented in other settings.

Biased statements

• The authors make several highly biased and politicized statements, such as the following: “The shift towards medicalization of childbirth resulted in a change in the perception of childbirth, transforming it from a natural process to a medical condition that necessitates intervention. Consequently, women are no longer regarded as authorities on their own bodies.” Many perinatal health professionals would disagree with these statements, and they are unnecessary politicized statements that are not required to support the benefit of the intervention under research.

• Similarly, the authors make statements about the supremacy of midwives (“A possible explanation for [higher satisfaction] is the care provided by midwives during labour and birth, and the physician was less involved.”) that serve only to create division rather than team-based care among health professionals.

• The authors state the findings are only relevant to midwives and midwifery students; why could nurses and physicians not implement this tool?

I hope that these comments are helpful as the authors consider revision to this article.

6. PLOS authors have the option to publish the peer review history of their article (what does this mean? ). If published, this will include your full peer review and any attached files.

**Do you want your identity to be public for this peer review?** For information about this choice, including consent withdrawal, please see our Privacy Policy .

Reviewer #1: No

Reviewer #2: No

Reviewer #3: No

---

## [Author Response · Author response to Decision Letter 1]

3 Oct 2024

Manuscript title: The effect of Labour Hopscotch Framework on maternal and neonatal outcomes in pregnant women: A randomized controlled trial

Dear Editor-in-chief:

Our team appreciates the editor and the reviewers for their observations and comments on the manuscript. We have fulfilled their comments and suggestions and wish to submit a revised version of the manuscript for further consideration in the journal. The changes in the revised version of the manuscript are marked as track changes. Below, we also provide a point-by-point response explaining how we have addressed each of the reviewer and editors’ comments.

Yours sincerely,

Dr. Parvin Abedi

Reply to the editor's comments

1-Editor's comment: Please ensure that your manuscript meets PLOS ONE's style requirements, including those for file naming.

Author's response: Thank you for your feedback. I will make sure that the manuscript complies with PLOS ONE's style requirements, including the proper file naming conventions, prior to resubmission.

2-Editor's comment: Thank you for stating the following financial disclosure: "This article constituted a segment of SA's PhD dissertation. We extend our thanks to the Research Deputy of Ahvaz Jundishapur University, faculty members of the Nursing and Midwifery School, and the management and birth ward staff at Shahid Baghaei Hospital for their invaluable assistance and support. Gratitude is also extended to all study participants for their willingness to take part. Ahvaz Jundishapur University of Medical Sciences is the founder of the study (grant code: RHPRC-0125.2023-01-28)." Please state what role the funders took in the study. If the funders had no role, please state: ""The funders had no role in study design, data collection and analysis, decision to publish, or preparation of the manuscript. "If this statement is not correct you must amend it as needed. Please include this amended Role of Funder statement in your cover letter; we will change the online submission form on your behalf.

Author's response: Thank you for your feedback regarding the financial disclosure. We have clarified the role of the funders in the study. Specifically, the funders provided support for the research but had no involvement in study design, data collection and analysis, decision to publish, or preparation of the manuscript. I will ensure this amended statement is included in the cover letter and will update the online submission form accordingly.

3-Editor's comment: Thank you for stating the following in the Acknowledgments Section of your manuscript. We note that you have provided funding information that is currently declared in your Funding Statement. However, funding information should not appear in the Acknowledgments section or other areas of your manuscript. We will only publish funding information present in the Funding Statement section of the online submission form. Please remove any funding-related text from the manuscript and let us know how you would like to update your Funding Statement. Currently, your Funding Statement reads as follows: "This article constituted a segment of SA's PhD dissertation. We extend our thanks to the Research Deputy of Ahvaz Jundishapur University, faculty members of the Nursing and Midwifery School, and the management and birth ward staff at Shahid Baghaei Hospital for their invaluable assistance and support. Gratitude is also extended to all study participants for their willingness to take part. Ahvaz Jundishapur University of Medical Sciences is the founder of the study (grant code: RHPRC-0125.2023-01-28)."Please include your amended statements within your cover letter; we will change the online submission form on your behalf.

Author's response: Thank you for your guidance regarding the funding information in my manuscript. I removed the funding details from the Acknowledgments section as you suggested.

4-Editor's comment: In the online submission form, you indicated that [The datasets generated and analyzed during the current study were available from the corresponding author on reasonable request.]. All PLOS journals now require all data underlying the findings described in their manuscript to be freely available to other researchers, either 1. In a public repository, 2. Within the manuscript itself, or 3. Uploaded as supplementary information. This policy applies to all data except where public deposition would breach compliance with the protocol approved by your research ethics board. If your data cannot be made publicly available for ethical or legal reasons (e.g., public availability would compromise patient privacy), please explain your reasons on resubmission and your exemption request will be escalated for approval.

Author's response: Thank you for your feedback. Due to ethical considerations and our commitment to protecting participant confidentiality, the datasets generated and analyzed during this study cannot be made publicly available. Our research ethics board requires strict confidentiality protocols to prevent any potential breaches of privacy. We are, however, willing to provide data on a case-by-case basis to researchers upon reasonable request, pending approval from our ethics board.

Suggestion, Question, or Comment from the reviewer 1 Author’s Response Change in the Manuscript (Yellow highlight)

We evaluated the impact of the LHF on maternal and neonatal outcomes. Between March 2023 and 22 October 2023, a randomized controlled trial including two parallel groups was carried out with 124 primiparous women with a term pregnancy (37 weeks or longer) in Ahvaz, Iran. Thank you for your feedback. it has been corrected It has been modified in the Abstract section: line 22

Women who were 18 years of age without cephalopelvic disproportion, with a low-risk singleton pregnancy, and who attended specialised childbirth education classes were among the eligible participants for the study. Thanks to the reviewer's opinion, it has been corrected It has been modified in the Abstract section: line 26

Women who were contraindicated for vaginal birth, were carrying an abnormal fetus, or had one or more medical disorders were excluded from the study. Thanks to the attention of the respected reviewer, it has been corrected It has been modified in the Abstract section: line 28

Women in the LHF group initiated breastfeeding earlier than the control group after birth Thanks for your comment, it has been corrected It has been modified in the Abstract section: line 34

The mean satisfaction score was higher among women using the LHF Thanks for your comment, it has been corrected It has been modified in the Abstract section: line 37

substantialrole (needs a space between words) Thanks for your comment, it has been corrected

Line 49 – ‘The care provided to women during childbirth has undergone significant changes during these years.’ What years are the authors referring to here? Thank you for your feedback regarding the clarity of the text. In light of the third reviewer's comment, we have decided to remove the sentence in line 49 to avoid any ambiguity.

Line 92 - Since there hasn't been any examination of LHF in Iran, it is possible to see LHF assists women in decision-making during labour as a strategic intervention. Could the authors please provide a better description of the LHF and consider using a figure to illustrate its use. Based on the description here I had difficulty understanding what it was until I looked up the original Carroll et al paper, which gave me a better idea of what the LHF is and how it is used. I don’t get this from your paper. Thanks to the opinion of the respected reviewer, we asked Carroll et al. to use the LHF figure, but we did not receive a response, and to protect the copyright of the original author, we only cited their article in the manuscript. Also, more explanation about LHF was added to the introduction. It has been modified in the Introduction section: line 66 to 84

Information about the Iran maternity system would also provide some context for your readers to help them better understand how women receive maternity care in Iran. For example, do women have choice in how they access maternity care, and are there different models of care available? If so, these could all impact how the LHF was received. Thank you for your insightful suggestion regarding the inclusion of information about the maternity care system in Iran. In the revised manuscript, I included details about the options women have for accessing maternity care and how these factors influence the reception of the LHF. It has been modified in the Setting section: line 124 to 136

Line 104 – Please explain why being married was part of the inclusion criteria. Thank you. This criterion was established to focus on a specific demographic, as marriage can influence women's access to maternity care and support systems in our study context. Understanding the experiences of married women allows us to explore dynamics related to family involvement and decision-making in maternity care.

Line 105 – how was cephalopelvic disproportion assessed? Thank you. In the revised manuscript, I provided a detailed explanation of the assessment process conducted by an obstetrician. This assessment was complemented by clinical evaluations of fetal size and, when necessary, ultrasound imaging to ensure accurate measurements. It has been modified in the Methods section: line 111

Line 104 – Please correct the English in this sentence. ‘Primiparous married women who were 18 years of age or older, gestational ages of 37 weeks or longer, plans for a vaginal birth, cephalopelvic proportion, low-risk singleton pregnancies, estimated fetal weights between 2500 and 4000 grams (According to the last ultrasound in the third trimester), basic literacy, and attendance at childbirth education classes were among the eligible participants for the study.’ Also, you should state that the childbirth classes were specific to the LHF. Thanks for your comment, it has been corrected It has been modified in the Methods section: line 109 to 120

Line 105 – ‘Although Shahid Baghai hospital is affiliated to the armed forces, it accepts all people.’ Is this hospital therefore a government run hospital? Please clarify as you later refer to government hospitals not offering epidurals or pharmacological pain relief. This will add context. Although Shahid Baghai Hospital is affiliated with the armed forces and serves all people, it does not offer services such as epidurals or pharmacological pain relief, similar to government hospitals. Therefore, it operates in a way that limits access to certain pain management options, which is an important context to consider. It has been modified in the Methods section: line 137

Line 122 – it is a risk to have a lay person obtaining informed consent. Please explain why the clinic secretary undertook this task. Thank you for your valuable comment. We would like to clarify that while the clinic secretary was tasked with distributing and collecting the consent forms, the responsibility for explaining the study and obtaining informed consent was solely carried out by the researcher. The researcher personally provided all necessary information to participants and answered their questions to ensure a thorough understanding of the study. This approach was adopted to facilitate the administrative aspects of the process without compromising the ethical requirements of informed consent. It has been modified in the Methods section: line 144 to 147

Line 145 – was the physiological childbirth workshop specific to the LHF? Please clarify. Thank you for your insightful question. The content of the physiological childbirth workshops was not exclusive to the Labour Hopscotch Framework (LHF), as the steps of the LHF were already embedded within the principles of physiological childbirth. However, what made the LHF particularly interesting for us was the way it structured these steps into a clear and progressive framework. This step-by-step approach provided a systematic method that we found valuable for further exploration and study. It has been modified in the Methods section: line 169

Line 149 – why was the TENS device excluded? Thank you for your observation. I would like to clarify that the TENS device was not excluded from the study. There was a misunderstanding due to the phrasing of the original sentence, which has now been corrected to reflect the accurate inclusion of TENS in the study. It has been modified in the Methods section: line 174

It is unclear if the researcher was the primary midwife supporting the woman in labour or if she was there in a support role. Please clarify. Thank you for your inquiry. The researcher functioned as the primary midwife, providing comprehensive care and support throughout the entire labor process, including the delivery itself, as well as postpartum care. This continuity of care ensured a consistent and supportive environment for the participants at all stages of their childbirth experience. It has been modified in the Methods section: line 186 to 189

Line 193 – please clarify why husband’s age was important demographic information to collect Thank you for your question. The collection of demographic information, including the husband’s age, is important as it provides valuable context for understanding the social and cultural factors influencing the participants' experiences. In the context of our research, examining the husband’s age allows for a more comprehensive analysis of family dynamics and potential correlations with childbirth outcomes. As this article is part of a larger thesis, gathering a wide range of demographic indicators is essential for establishing a nuanced understanding of the population studied. It has been modified in the Methods section: line 234 to 237

Line 227 – please provide comment on the Cronbach’s alpha, whether this is considered reliable. Thanks for your comment

Yes, it is reliable and the sentence is complete in the text It has been modified in the Methods section: line 267

Line 237 – ‘Due to the fact that one person from each group had a drop’. Please explain what a drop is? Or do you mean one person from each group dropped out? Thank you for your inquiry regarding the terminology used. Corrected. It has been modified in the Results section: line 297

Lines 263 – 265 – Measuring age and employment status of husbands/partners is something that is rarely done in English-speaking countries unless the study is specific to fathers so it would be good to provide some context around why this was important to include in your study. These potentially are cultural nuances that might not be understood outside of Iran. Thank you for your question. In our study, measuring the age and employment status of husbands was essential due to the cultural and socio-economic context in Iran, where the role of the husband/partner is often strongly tied to family decision-making, particularly in areas such as healthcare access, financial support, and childbirth decisions. Employment status is a key indicator of economic stability, which can directly influence a family's ability to seek and maintain adequate prenatal and postnatal care. Additionally, in many Iranian families, the age and employment status of the husband are associated with their level of involvement, support, and readiness for childbirth. These factors can significantly impact maternal outcomes, which is why they were included in the study. Understanding these dynamics helps provide a more complete picture of maternal and family health within this specific cultural context. It has been modified in the Methods section: line 234 to 237

Line 303 – You can’t say that the LHF has increased the normal vaginal birthrate in Iran. All you can say is that using the LHF increased the NVB rate in the study sample. Thank you for your insightful feedback. I revised the manuscript to ensure clarity and avoid overgeneralization to the broader population in Iran. It has been modified in the Discussion section: line 364 to 366

Line 339 – ‘Supporting pregnant women and making them participate in the labor process.’ I think you can facilitate women to participate in the labour process but not make them. Please reconsider this sentence.

---

## [Decision Letter · Decision Letter 1]

11 Nov 2024

PONE-D-24-31506R1The effect of Labour Hopscotch Framework on maternal and neonatal outcomes in pregnant women: A randomized controlled trialPLOS ONE

Dear Dr. Abedi,

Thank you for submitting your manuscript to PLOS ONE. After careful consideration, we feel that it has merit but does not fully meet PLOS ONE’s publication criteria as it currently stands. Therefore, we invite you to submit a revised version of the manuscript that addresses the points raised during the review process.

**ACADEMIC EDITOR: **
**Please respond to all reviewers comments.**

We look forward to receiving your revised manuscript.

Kind regards,

Ahmed Mohamed Maged, MD

Academic Editor

PLOS ONE

Reviewers' comments:

Reviewer's Responses to Questions

**Comments to the Author**

1. If the authors have adequately addressed your comments raised in a previous round of review and you feel that this manuscript is now acceptable for publication, you may indicate that here to bypass the “Comments to the Author” section, enter your conflict of interest statement in the “Confidential to Editor” section, and submit your "Accept" recommendation.

Reviewer #1: All comments have been addressed

Reviewer #4: All comments have been addressed

2. Is the manuscript technically sound, and do the data support the conclusions?

Reviewer #1: Yes

Reviewer #4: Yes

3. Has the statistical analysis been performed appropriately and rigorously? 

Reviewer #1: Yes

Reviewer #4: Yes

4. Have the authors made all data underlying the findings in their manuscript fully available?

Reviewer #1: No

Reviewer #4: No

5. Is the manuscript presented in an intelligible fashion and written in standard English?

Reviewer #1: No

Reviewer #4: Yes

6. Review Comments to the Author

Reviewer #1: Thank you for the opportunity to review this paper again after the revision. The authors have done a great job at addressing the issues with the first submission and it is a much better read now. I have a few comments for consideration. Your paper would still benefit from an edit for use of English as there are still many places that don’t quite make sense to a native English speaker, or are clumsily worded. In addition, the use of ‘birth’ instead of ‘delivery’ is more woman-centred, so please consider changing this throughout the manuscript. I have some futher minor considerations below that need to be addressed before publication:

Line 32 – women who used the LHF (and again throughout the manuscript)

Line 33 – women using the LHF were 10.3 times more likely to experience a vaginal birth

Line 36 – no difference between the two groups for exclusive breastfeeding

Line 60 – recent study conducted there

Line 99 – the research aimed to…

Line 137 – labour and birth are supported by midwives (not performed)

Line 149 – your software should be referenced

Line 175 – as you have explained the seven steps of the LHF, you don’t need to repeat them here

Line 205 – you have abbreviated caesarean section to c/s here but have used C-section elsewhere. Please be consistant.

Line 204-218 – extensive detailing of what a first, second and third degree tear are, as well as explaining Apgar scoring, is not required as these are universal measures

Line 270 – please reference your software

Line 293 – eligibile women that were recruited

Line 309 – women who used the LHF

Line 312 – women who used the LHF

Line 322 – please revise the sentence beginning with ‘as evident in table 3’ as this doesn’t quite make sense

Line 334 – scored above what?

Line 342 – what do the authors mean by ‘satisfaction with the baby’?

Line 350 – you don’t need to introduce your acronym again as this was done previously in the paper

Line 374 – correlation between using the LHF

Line 385 – again, you don’t need to introduce your acronym

Line 428 – women using the LHF

Line 445 – I’m not sure how a focus on women’s health will reduce the cesaerean section rate. A focus on midwives facilitating physiological birth would help but as women’s health is such a braod subject, you need to rephrase this.

Line 451 – this phrase needs rewording as it doesn’t make sense in English. In fact, the whole limitations section needs re-working as it is difficult to understand, and the English is not correct.

Line 458 – increases the initiation of breastfeeding

Figure 1 – In the follow-up box for the control group, you have ‘state reason for dropping out’. I don’t think that is meant to be there.

Table 4 – length is used rather than height as babies can’t stand up to be measured

Reviewer #4: Thank you for the opportunity to review the revised version of this manuscript regarding the LHF intervention for increasing physiologic birth, length of labor, breastfeeding, morbidity markers and satisfaction. The authors have been responsive to prior critiques.

The manuscript will be stronger through addressing a few additional issues:

1. Agree with the prior reviewer that it would be ideal to include a figure or graphic that communicates the LHF intervention. One option that the authors might explore with the journal would be to include Carroll et al's figure and cite their work. This will likely require permission from Carroll and/or the journal that published the original work but it would be worth pursuing

2. Methods- pelvic examination prior to labor as a means to assess for cephalopelvic proportion is not an evidence based way of assessing the potential fit of the fetuses' head through the pelvis. Is this a common practice in Iran? Recommend citing this approach and/or adding more comment about this part of the inclusion criteria.

3. Methods- Two areas of potential bias should be noted in the limitations section of this study. a) Those who were randomized to the intervention were provided with the researcher's phone numbers (vs those randomized to usual care)- this may have increased perceptions of support among those in the intervention arm, B) and given that the midwife attending the labor and births of all in the study is the researcher, the attending midwife's knowledge of who was and was not in the LHF groups may have unconsciously biased the care they received.

4. Maternal outcomes- Some readers will be confused about the negative effect sizes. Consider adding text that helps the reader interpret these findings by communicating the results in terms of both effect size and actual time differences in length of labor

7. PLOS authors have the option to publish the peer review history of their article (what does this mean? ). If published, this will include your full peer review and any attached files.

**Do you want your identity to be public for this peer review?** For information about this choice, including consent withdrawal, please see our Privacy Policy .

Reviewer #1: No

Reviewer #4: No

---

## [Author Response · Author response to Decision Letter 2]

15 Nov 2024

Manuscript title: The effect of Labour Hopscotch Framework on maternal and neonatal outcomes in pregnant women: A randomized controlled trial

Dear Editor-in-chief:

Our team appreciates the editor and the reviewers for their observations and comments on the manuscript. We have fulfilled their comments and suggestions and wish to submit a revised version of the manuscript for further consideration in the journal. The changes in the revised version of the manuscript are marked as track changes. Below, we also provide a point-by-point response explaining how we have addressed each of the reviewer' comments.

Yours sincerely,

Dr. Parvin Abedi

Suggestion, Question, or Comment from the reviewer 1 Author’s Response Change in the Manuscript (Yellow highlight)

the use of ‘birth’ instead of ‘delivery’ is more woman-centered, so please consider changing this throughout the manuscript. Thanks for your comment, it has been corrected Applied to all text

Line 32 – women who used the LHF (and again throughout the manuscript)

Line 33 – women using the LHF were 10.3 times more likely to experience a vaginal birth

Line 36 – no difference between the two groups for exclusive breastfeeding

Line 60 – recent study conducted there

Line 99 – the research aimed to…

Line 137 – labour and birth are supported by midwives (not performed) Thank you for your feedback. it has been corrected Line 31

Line 32

Line 34

Line 59

Line 98

Line 139

Line 149 – your software should be referenced Thanks to the reviewer's opinion, it has been corrected Line 151

Line 175 – as you have explained the seven steps of the LHF, you don’t need to repeat them here Thanks to the attention of the respected reviewer, it has been corrected Line 177

Line 205 – you have abbreviated caesarean section to c/s here but have used C-section elsewhere. Please be consistant. Thanks for your comment, it has been corrected Applied to all text

Line 204-218 – extensive detailing of what a first, second and third degree tear are, as well as explaining Apgar scoring, is not required as these are universal measures Thanks for your comment, it has been corrected Line 210

Line 270 – please reference your software Thanks for your comment, it has been corrected Line 264

Line 293 – eligibile women that were recruited

Line 309 – women who used the LHF

Line 312 – women who used the LHF Thanks for your comment, it has been corrected Line 287

Line 303

Line 306

Line 322 – please revise the sentence beginning with ‘as evident in table 3’ as this doesn’t quite make sense Thanks for your comment, it has been corrected Line 322

Line 334 – scored above what? Thanks for your comment, it has been corrected. Descriptions were added to the text Line 334

Line 342 – what do the authors mean by ‘satisfaction with the baby’? Thanks for your comment, satisfaction with the baby" refers to the mother's emotional and psychological feelings regarding her newborn after birth, encompassing aspects like bonding, perceived well-being, and overall happiness with the baby’s health and appearance. This measure aims to capture the mother's sense of fulfillment and attachment to the child in the early postpartum period.

Line 350 – you don’t need to introduce your acronym again as this was done previously in the paper Thanks for your comment, it has been corrected Line 350

Line 374 – correlation between using the LHF

Line 385 – again, you don’t need to introduce your acronym

Line 428 – women using the LHF Thanks for your comment, it has been corrected Line 372

Line 383

Line 426

Line 445 – I’m not sure how a focus on women’s health will reduce the cesaerean section rate. A focus on midwives facilitating physiological birth would help but as women’s health is such a braod subject, you need to rephrase this. Thanks for your comment, it has been corrected Line 442

Line 451 – this phrase needs rewording as it doesn’t make sense in English. In fact, the whole limitations section needs re-working as it is difficult to understand, and the English is not correct. Thanks for your comment, it has been corrected Line 447

Line 458 – increases the initiation of breastfeeding

Figure 1 – In the follow-up box for the control group, you have ‘state reason for dropping out’. I don’t think that is meant to be there.

Table 4 – length is used rather than height as babies can’t stand up to be measured Thanks for your comment, it has been corrected Line 456

Figure 1

Table 4

Suggestion, Question, or Comment from the reviewer 4 Author’s Response Change in the Manuscript (blue highlight)

1. Agree with the prior reviewer that it would be ideal to include a figure or graphic that communicates the LHF intervention. One option that the authors might explore with the journal would be to include Carroll et al's figure and cite their work. This will likely require permission from Carroll and/or the journal that published the original work but it would be worth pursuing Thanks to the opinion of the respected reviewer, we asked Carroll et al. to use the LHF figure, but we did not receive a response, and to protect the copyright of the original author, we only cited their article in the manuscript. Also, more explanation about LHF was added to the introduction.

2. Methods- pelvic examination prior to labor as a means to assess for cephalopelvic proportion is not an evidence based way of assessing the potential fit of the fetuses' head through the pelvis. Is this a common practice in Iran? Recommend citing this approach and/or adding more comment about this part of the inclusion criteria. Thanks to the reviewer's opinion, In Iran, pelvic examination prior to labor is commonly used as a method to assess the fit of the fetal head with the mother's pelvis. Although the scientific evidence regarding the accuracy of this method is mixed, it remains widely utilized in many healthcare settings due to its simplicity and ease of access. Line 109

3. Methods- Two areas of potential bias should be noted in the limitations section of this study. a) Those who were randomized to the intervention were provided with the researcher's phone numbers (vs those randomized to usual care)- this may have increased perceptions of support among those in the intervention arm, B) and given that the midwife attending the labor and births of all in the study is the researcher, the attending midwife's knowledge of who was and was not in the LHF groups may have unconsciously biased the care they received. Thank you for your valuable feedback. Regarding the first point, the researcher's phone number was provided to all participants, including those in the control group, to allow them to reach out for any necessary clarifications or support. This approach ensured equitable access to the researcher across both groups and aimed to prevent discrepancies in perceived support.

Regarding the second point, we acknowledge the limitation that neither the researcher nor the participants could be blinded due to the nature of the intervention. While every effort was made to provide consistent and impartial care, we recognize that the midwife's awareness of group assignments may have unintentionally influenced care. This potential bias has been transparently addressed in the limitations section of the manuscript.

Line 447 to 454

4. Maternal outcomes- Some readers will be confused about the negative effect sizes. Consider adding text that helps the reader interpret these findings by communicating the results in terms of both effect size and actual time differences in length of labor Thank you for your feedback. We have clarified the interpretation of negative effect sizes in the revised manuscript by linking them to shorter labor durations in the intervention group. Additionally, we have included the corresponding time differences to make the findings more accessible to readers. We hope this resolves any confusion. Line 313 to 319

---

## [Decision Letter · Decision Letter 2]

9 Dec 2024

PONE-D-24-31506R2The effect of Labour Hopscotch Framework on maternal and neonatal outcomes in pregnant women: A randomized controlled trialPLOS ONE

Dear Dr. Abedi,

Thank you for submitting your manuscript to PLOS ONE. After careful consideration, we feel that it has merit but does not fully meet PLOS ONE’s publication criteria as it currently stands. Therefore, we invite you to submit a revised version of the manuscript that addresses the points raised during the review process.

**ACADEMIC EDITOR: Please respond to all reviewers comments**==============================

We look forward to receiving your revised manuscript.

Kind regards,

Ahmed Mohamed Maged, MD

Academic Editor

PLOS ONE

Journal Requirements:

Reviewers' comments:

Reviewer's Responses to Questions

**Comments to the Author**

1. If the authors have adequately addressed your comments raised in a previous round of review and you feel that this manuscript is now acceptable for publication, you may indicate that here to bypass the “Comments to the Author” section, enter your conflict of interest statement in the “Confidential to Editor” section, and submit your "Accept" recommendation.

Reviewer #2: (No Response)

2. Is the manuscript technically sound, and do the data support the conclusions?

Reviewer #2: Yes

3. Has the statistical analysis been performed appropriately and rigorously? 

Reviewer #2: Yes

4. Have the authors made all data underlying the findings in their manuscript fully available?

Reviewer #2: No

5. Is the manuscript presented in an intelligible fashion and written in standard English?

Reviewer #2: Yes

6. Review Comments to the Author

Reviewer #2: The manuscript is very much improved since my last review. There are still a few pending issues with statistical methods and results which I would recommend that the authors review.

Abstract

- line 32 - please specify the units for the difference: is this hours, days, etc?

- line 33, 34 - please indicate the difference, with 95%CI in addition to the p-value, as done with the other outcome in this paragraph. Also state the units

Data analysis

- lines 266 to 268 - it is not clear what the objective of the chi-squared, Fisher's exact and t-tests described here were. In any case, there is no role for tests comparing groups in a randomised trial except to compare primary and secondary outcomes between intervention and control groups. My recommendation would be to drop these tests from the methods and results. Simple descriptive analyses reporting means and standard deviations or medians and interquartile ranges for continuous outcomes, and counts and proportions for categorical ones, should be sufficient. These should all be reported in table 1 and table 2 (this is probably already done so no further action except for removing any description and results pertaining to the unnecessary statistical tests described above).

Results

- following the last comment under 'data analysis' above, there should be no p-values for any tests comparing groups here and in the respective tables. Removing the p-values from Tables 1, 2 and 4 and from the main text referring to or interpreting these p-values (and amending the statistical methods as described above) will address this comment. Note that the p-values in Tables 3 and 5 are fine. However, for Table 5, you should be reporting standard errors (SE) not standard deviations (SD) - SDs are descriptive while SEs are used for inference. Since Table 5 is making inference on an outcome, SEs should be reported here. The rest of the table is fine.

- when reporting the results pertaining to outcomes, you should report means and standard errors for continuous outcomes in each group, followed by the difference with 95%CI (and p-value if desired); for binary outcomes, report counts and proportions in each groups followed by the odds ratio (or risk ratio or risk difference if preferred) with 95%CI (and p-value if desired). Some of this has already been done in the current version; I am not necessarily asking that you report p-values where you have not done so, if you have reported the 95%CI.

7. PLOS authors have the option to publish the peer review history of their article (what does this mean? ). If published, this will include your full peer review and any attached files.

**Do you want your identity to be public for this peer review?** For information about this choice, including consent withdrawal, please see our Privacy Policy .

Reviewer #2: No

---

## [Author Response · Author response to Decision Letter 3]

13 Dec 2024

Manuscript title: The effect of Labour Hopscotch Framework on maternal and neonatal outcomes in pregnant women: A randomized controlled trial

Dear Editor-in-chief:

Our team appreciates the editor and the reviewers for their observations and comments on the manuscript. We have fulfilled their comments and suggestions and wish to submit a revised version of the manuscript for further consideration in the journal. The changes in the revised version of the manuscript are marked as track changes. Below, we also provide a point-by-point response explaining how we have addressed each of the reviewer' comments.

Yours sincerely,

Dr. Parvin Abedi

Journal Requirements:

Author’s Response: Thank you for your suggestion. References 12 and 14 were identical, as were references 19 and 39. Therefore, both duplicates were removed from the reference list. Additionally, all references have been reviewed, and none of them have been retracted. Therefore, there are no issues regarding the retracted status of any references.

Suggestion, Question, or Comment from the reviewer 2 Author’s Response Change in the Manuscript (Yellow highlight)

- line 32 - please specify the units for the difference: is this hours, days, etc? Thank you for your comment. The units for the difference mentioned in line 32 are minutes. We have updated the manuscript to clarify this. Applied to all text

- line 33, 34 - please indicate the difference, with 95%CI in addition to the p-value, as done with the other outcome in this paragraph. Also state the units Thank you for your comment. We have added the 95% CI and clarified the units as minutes in lines 33 and 34. Line 33-34

- lines 266 to 268 - it is not clear what the objective of the chi-squared, Fisher's exact and t-tests described here were. In any case, there is no role for tests comparing groups in a randomised trial except to compare primary and secondary outcomes between intervention and control groups. My recommendation would be to drop these tests from the methods and results. Simple descriptive analyses reporting means and standard deviations or medians and interquartile ranges for continuous outcomes, and counts and proportions for categorical ones, should be sufficient. These should all be reported in table 1 and table 2 (this is probably already done so no further action except for removing any description and results pertaining to the unnecessary statistical tests described above). Thank you for your feedback. Following your recommendation, we have removed the chi-square, Fisher’s exact, and t-tests from the methods and results sections. We have now used simple descriptive analyses, reporting means and standard deviations or medians and interquartile ranges for continuous outcomes, and counts and proportions for categorical outcomes. These details are now included in the revised Table 1, Table 2, and Table 4. Table 1 and Table 2

- following the last comment under 'data analysis' above, there should be no p-values for any tests comparing groups here and in the respective tables. Removing the p-values from Tables 1, 2 and 4 and from the main text referring to or interpreting these p-values (and amending the statistical methods as described above) will address this comment. Note that the p-values in Tables 3 and 5 are fine. However, for Table 5, you should be reporting standard errors (SE) not standard deviations (SD) - SDs are descriptive while SEs are used for inference. Since Table 5 is making inference on an outcome, SEs should be reported here. The rest of the table is fine. Thank you for your comment. We have removed the p-values from Tables 1, 2, and 4, as well as from the main text referring to or interpreting these p-values, in line with your suggestions. We have also amended the statistical methods accordingly. Regarding Table 5, we have replaced the standard deviations (SD) with standard errors (SE), as SE is appropriate for inference in this context. The rest of the table remained unchanged. Table 1, 2 and Table 4

Table 5

- when reporting the results pertaining to outcomes, you should report means and standard errors for continuous outcomes in each group, followed by the difference with 95%CI (and p-value if desired); for binary outcomes, report counts and proportions in each groups followed by the odds ratio (or risk ratio or risk difference if preferred) with 95%CI (and p-value if desired). Some of this has already been done in the current version; I am not necessarily asking that you report p-values where you have not done so, if you have reported the 95%CI. Thank you for your comment. In response, we have now updated the results section to report means and standard errors for continuous outcomes in each group, followed by the difference with 95% CI, and p-values where appropriate. For binary outcomes, we have included counts and proportions in each group, followed by the odds ratio (or risk ratio/risk difference), along with the 95% CI and p-values where applicable. We have made sure to included this for all outcomes where appropriate, as per your guidance. Applied to all text

---

## [Decision Letter · Decision Letter 3]

30 Dec 2024

PONE-D-24-31506R3The effect of Labour Hopscotch Framework on maternal and neonatal outcomes in pregnant women: A randomized controlled trialPLOS ONE

Dear Dr. Abedi,

Thank you for submitting your manuscript to PLOS ONE. After careful consideration, we feel that it has merit but does not fully meet PLOS ONE’s publication criteria as it currently stands. Therefore, we invite you to submit a revised version of the manuscript that addresses the points raised during the review process.

ACADEMIC EDITOR: Please respond to all reviewer comments

We look forward to receiving your revised manuscript.

Kind regards,

Ahmed Mohamed Maged, MD

Academic Editor

PLOS ONE

Journal Requirements:

Reviewers' comments:

Reviewer's Responses to Questions

**Comments to the Author**

1. If the authors have adequately addressed your comments raised in a previous round of review and you feel that this manuscript is now acceptable for publication, you may indicate that here to bypass the “Comments to the Author” section, enter your conflict of interest statement in the “Confidential to Editor” section, and submit your "Accept" recommendation.

Reviewer #2: (No Response)

2. Is the manuscript technically sound, and do the data support the conclusions?

Reviewer #2: Yes

3. Has the statistical analysis been performed appropriately and rigorously? 

Reviewer #2: Yes

4. Have the authors made all data underlying the findings in their manuscript fully available?

Reviewer #2: No

5. Is the manuscript presented in an intelligible fashion and written in standard English?

Reviewer #2: Yes

6. Review Comments to the Author

Reviewer #2: I thank the author for responding positively to my previous comments. Unfortunately the recent round of edits has introduced a few new small problems which need to be addressed.

Lines 266 to 268 should read: "Categorical variables were summarised as counts and percentages, and continuous variables were summarised using means, standard deviation and ranges."

In line 268 and throughout the manuscript, there is a reference to 'effect sizes'; this is not quite correct, as this terminology has a very specific meaning in statistics, usually standardised estimates of effect, which is not what is meant here. For example, even though this statement is referring to mean differences and odds ratios, these are not standardised estimates. Throughout the manuscript, 'effect size' should be replaced with 'estimate of effect' EXCEPT (this is very important) where you are referring to Cohen's d, which are indeed standardised differences therefore true effect sizes. For example in line 268 and 270 'effect sizes' should be replaced with 'estimates of effect'.

The sentence on line 270 which includes the phrase '95% confidence level' should instead say '95% confidence interval.' and end there (i.e. '... used to ensure statistical robustness' should be removed). Alternatively (and preferably), the whole sentence that begins in line 269 with 'confidence intervals' should be dropped and the line on line 268 which begins with 'effect sizes' should be edited as recommended above and to include 95% confidence intervals, i.e. "Estimates of effect with 95% confidence intervals were calculated to assess the magnitude...". Additionally, the last sentence that begins on line 273 and ends on 274 is unnecessary (because this information will have been reported above and inferred in the sample size calculation) and should therefore be dropped.

The sentence on line 271 about one dropout is a result and should be removed. You should simply say that the analysis was per-protocol, and leave it at that.

It is unclear what is meant by ± in various places in the text, for example, lines 292, 293, 303, 331, and 332. You should instead very clearly indicate what these quantities are. For example if they are SDs in line 292, you should say: "Women in the intervention group had an average age of 22.4 (SD 4.5) years, slightly lower than the control group's average of 22.6 (SD 4.8) years." Please make this change wherever you have used ± in the text of the manuscript (note that this seems to be fine in the tables as you have explained what you mean there).

Specifically for the use of ± in the main text of the section titled "maternal outcomes", since these are outcomes, then the quantities preceded by ± should be standard errors. Please ensure that these are in fact standard errors not standard deviations, and report them as suggested above (and also indicate whether the quantities reported are means), for example: "Women who used the LHF experienced shorter first, second, and third stages of labor and shorter total labor duration in contrast to the control group (mean 162.6, SE 55min vs. mean 260.30, SE 95.3 min)". The same should apply to childbirth satisfaction and positive experience on line 331 and 332 (i.e. indicate if the quantities are means and check that you are reporting the SEs as shown above) since these are outcomes.

Apologies if I have raised some points not previously raised; as the manuscript has improved, some of the reporting has become clearer to me making it easier to spot elements that seemed unusual.

7. PLOS authors have the option to publish the peer review history of their article (what does this mean? ). If published, this will include your full peer review and any attached files.

**Do you want your identity to be public for this peer review?** For information about this choice, including consent withdrawal, please see our Privacy Policy .

Reviewer #2: No

---

## [Author Response · Author response to Decision Letter 4]

2 Jan 2025

Manuscript title: The effect of Labour Hopscotch Framework on maternal and neonatal outcomes in pregnant women: A randomized controlled trial

Dear Editor-in-chief:

Our team appreciates the editor and the reviewers for their observations and comments on the manuscript. We have fulfilled their comments and suggestions and wish to submit a revised version of the manuscript for further consideration in the journal. The changes in the revised version of the manuscript are marked as track changes. Below, we also provide a point-by-point response explaining how we have addressed each of the reviewer' comments.

Yours sincerely,

Dr. Parvin Abedi

Suggestion, Question, or Comment from the reviewer 2 Author’s Response Change in the Manuscript (Yellow highlight)

Lines 266 to 268 should read: "Categorical variables were summarised as counts and percentages, and continuous variables were summarised using means, standard deviation and ranges." Thank you for your suggestion. We have revised lines 266 to 268 as recommended to improve clarity and precision. Lines 266 to 268

In line 268 and throughout the manuscript, there is a reference to 'effect sizes'; this is not quite correct, as this terminology has a very specific meaning in statistics, usually standardised estimates of effect, which is not what is meant here. For example, even though this statement is referring to mean differences and odds ratios, these are not standardised estimates. Throughout the manuscript, 'effect size' should be replaced with 'estimate of effect' EXCEPT (this is very important) where you are referring to Cohen's d, which are indeed standardised differences therefore true effect sizes. For example in line 268 and 270 'effect sizes' should be replaced with 'estimates of effect'. Thank you for your careful reading and constructive feedback.

I would like to clarify that in this study, we used Cohen's d as the standardized effect size, which is correctly referred to as a "true effect size" in statistical terminology. Therefore, the use of the term "effect size" throughout the manuscript is appropriate, and there was no intention to replace it with "estimate of effect."

Regarding lines 268 and 270 where "effect sizes" are mentioned, it is important to note that we are referring to Cohen's d, which is indeed a standardized difference and a true effect size. I revised the due sentences for better clarification.

Lines 268 to 270

The sentence on line 270 which includes the phrase '95% confidence level' should instead say '95% confidence interval.' and end there (i.e. '... used to ensure statistical robustness' should be removed). Alternatively, (and preferably), the whole sentence that begins in line 269 with 'confidence intervals' should be dropped and the line on line 268 which begins with 'effect sizes' should be edited as recommended above and to include 95% confidence intervals, i.e. "Estimates of effect with 95% confidence intervals were calculated to assess the magnitude...". Additionally, the last sentence that begins on line 273 and ends on 274 is unnecessary (because this information will have been reported above and inferred in the sample size calculation) and should therefore be dropped. Thank you for your feedback. We revised line 270 to "95% confidence interval" and updated line 268 to include "Estimates of effect with 95% confidence intervals." The redundant sentence on line 273 was also removed as suggested. Lines 266 to 270

The sentence on line 271 about one dropout is a result and should be removed. You should simply say that the analysis was per-protocol, and leave it at that. Thank you for your suggestion. We have removed the sentence about the dropout on line 271 and revised the text to indicate that the analysis was conducted per protocol, as recommended. Line 270

It is unclear what is meant by ± in various places in the text, for example, lines 292, 293, 303, 331, and 332. You should instead very clearly indicate what these quantities are. For example, if they are SDs in line 292, you should say: "Women in the intervention group had an average age of 22.4 (SD 4.5) years, slightly lower than the control group's average of 22.6 (SD 4.8) years." Please make this change wherever you have used ± in the text of the manuscript (note that this seems to be fine in the tables as you have explained what you mean there). Thank you for your comment. We have replaced the ± symbols in the text with clear descriptions, specifying whether the values represent standard deviations (SDs) or other measures, as suggested. Line 289 to 290

Specifically, for the use of ± in the main text of the section titled "maternal outcomes", since these are outcomes, then the quantities preceded by ± should be standard errors. Please ensure that these are in fact standard errors not standard deviations, and report them as suggested above (and also indicate whether the quantities reported are means), for example: "Women who used the LHF experienced shorter first, second, and third stages of labor and shorter total labor duration in contrast to the control group (mean 162.6, SE 55min vs. mean 260.30, SE 95.3 min)". The same should apply to childbirth satisfaction and positive experience on line 331 and 332 (i.e. indicate if the quantities are means and check that you are reporting the SEs as shown above) since these are outcomes. Thank you for your comment. We have ensured that the quantities preceded by ± in the "maternal outcomes" section are standard errors (SEs) and not standard deviations. These have been clearly reported as SEs along with the means, as per your suggestion. Line 302 to 303 and line 329 to 330

Apologies if I have raised some points not previously raised; as the manuscript has improved, some of the reporting has become clearer to me making it easier to spot elements that seemed unusual. Thank you for your thoughtful review and kind words about the improvements in the manuscript. We appreciate your continued attention to detail, which helps us further refine and enhance the clarity and quality of our work.

---

## [Decision Letter · Decision Letter 4]

28 Jan 2025

The effect of Labour Hopscotch Framework on maternal and neonatal outcomes in pregnant women: A randomized controlled trial

PONE-D-24-31506R4

Dear Dr. Abedi,

We’re pleased to inform you that your manuscript has been judged scientifically suitable for publication and will be formally accepted for publication once it meets all outstanding technical requirements.

Kind regards,

Ahmed Mohamed Maged, MD

Academic Editor

PLOS ONE

Additional Editor Comments (optional):

Reviewers' comments:

Reviewer's Responses to Questions

**Comments to the Author**

1. If the authors have adequately addressed your comments raised in a previous round of review and you feel that this manuscript is now acceptable for publication, you may indicate that here to bypass the “Comments to the Author” section, enter your conflict of interest statement in the “Confidential to Editor” section, and submit your "Accept" recommendation.

Reviewer #2: All comments have been addressed

2. Is the manuscript technically sound, and do the data support the conclusions?

Reviewer #2: (No Response)

3. Has the statistical analysis been performed appropriately and rigorously? 

Reviewer #2: (No Response)

4. Have the authors made all data underlying the findings in their manuscript fully available?

Reviewer #2: (No Response)

5. Is the manuscript presented in an intelligible fashion and written in standard English?

Reviewer #2: (No Response)

6. Review Comments to the Author

Reviewer #2: Please edit line 278 to read: "The analysis was done per-protocol."

Describing it as a method is not quite right.

7. PLOS authors have the option to publish the peer review history of their article (what does this mean? ). If published, this will include your full peer review and any attached files.

**Do you want your identity to be public for this peer review?** For information about this choice, including consent withdrawal, please see our Privacy Policy .

Reviewer #2: No

---

## [Editor Report · Acceptance letter]

PONE-D-24-31506R4

PLOS ONE

Dear Dr. Abedi,

I'm pleased to inform you that your manuscript has been deemed suitable for publication in PLOS ONE. Congratulations! Your manuscript is now being handed over to our production team.

Kind regards,

on behalf of

Professor Ahmed Mohamed Maged

Academic Editor

PLOS ONE